# Atlantic origin of the increasing Asian westerly jet interannual variability

Lifei Lin [1,2,12], Chundi Hu [1,12] ✉, Bin Wang [3], Renguang Wu [4], Zeming Wu [1], Song Yang [2,5], Wenju Cai [6,7,8,9], Peiliang Li[1], Xuejun Xiong[1] & Dake Chen[2,10,11]

The summer Eurasian westerly jet is reported to become weaker and wavier, thus promoting the frequent weather extremes. However, the primary driver of the changing jet stream remains in debate, mainly due to the regionality and seasonality of the Eurasian jet. Here we report a sharp increase, by approximately 140%, in the interannual variability of the summertime East Asian jet (EAJ) since the end of twentieth century. Such interdecadal change induces considerable changes in the large-scale circulation pattern across Eurasia, and consequently weather and climate extremes including heatwaves, droughts, and Asian monsoonal rainfall regime shifts. The trigger mainly emerges from preceding February North Atlantic seesaw called Scandinavian pattern (contributing to $81.1 \pm 2.9\%$ of the enhanced EAJ variability), which harnesses the "cross-seasonal-coupled oceanic-atmospheric bridge" to exert a delayed impact on EAJ and thus aids relevant predictions five months in advance. However, projections from state-of-the-art models with prescribed anthropogenic forcing exhibit no similar circulation changes. This sheds light on that, at the interannual timescale, a substantial portion of recently increasing variability in the East Asian sector of the Eurasian westerly jet arises from unforced natural variability.

The upper-level westerly jet, prevailing in the mid- to high-latitudes of Earth's atmosphere, "steers" the movement of air masses and frontal zones, thus exerting severe impacts on global weather and climate[1–3], strongly threatening ecosystem[4], food security[5] and human health[6]. Its variability can be conceptually viewed as a pulsing in strength, shifting in latitude or change in waviness[1]. Bouts of weather extremes, including heatwaves, droughts, flood-producing storms and wildfires, have been linked to abnormal jet stream[6–9]. Summer 2020 saw an extreme

rainfall over the middle and lower reaches of the Yangtze River Valley, triggered by a record-breaking Asian subtropical jet stream in its intensity[10–12]. The more-persistent double jets configuration over Eurasia is reported as the culprit of accelerated heatwave trends over western European[13]. In terms of the regional ecosystem such as radial tree growth[4], a southwestward shifted jet stream over the North Atlantic-Europe results in a reduction of 38% in radial tree growth over southeastern Europe. On the issue of food security, a strongly

---

[1]Ocean College, Zhejiang University, Zhoushan, China. [2]School of Atmospheric Sciences, Sun Yat-sen University; and Southern Marine Science and Engineering Guangdong Laboratory (Zhuhai), Zhuhai, China. [3]Department of Atmospheric Sciences and International Pacific Research Center, School of Ocean Earth Science and Technology, University of Hawaii at Manoa, Honolulu, HI, USA. [4]School of Earth Sciences, Zhejiang University, Hangzhou, China. [5]Guangdong Province Key Laboratory for Climate Change and Natural Disaster Studies, Sun Yat-sen University, Zhuhai, China. [6]Frontiers Science Center for Deep Ocean Multispheres and Earth System/Physical Oceanography Laboratory/Sanya Oceanographic Institution, Ocean University of China, Qingdao, China. [7]Laoshan Laboratory, Qingdao, China. [8]State Key Laboratory of Loess and Quaternary Geology, Institute of Earth Environment, Chinese Academy of Sciences, Xi'an, China. [9]State Key Laboratory of Marine Environmental Science & College of Ocean and Earth Sciences, Xiamen University, Xiamen, China. [10]State Key Laboratory of Satellite Ocean Environment DynamicsSecond Institute of Oceanography, Ministry of Natural Resources, Hangzhou, China. [11]School of Oceanography, Shanghai Jiao Tong University, Shanghai, China. [12]These authors contributed equally: Lifei Lin, Chundi Hu. ✉e-mail: hucd@zju.edu.cn

meandering jet stream is capable of triggering simultaneous harvest failures over major crop-producing regions[5]. In aviation, the takeoff performance, aviation safety, optimal flight route and consequent flight time are substantially influenced by weather extremes and shear-driven clear-air turbulence related to abnormal jet stream[14].

In the warming climate, the Eurasian mid-latitude circulation is reported to behave in a wavier manner during summer[15,16], concurrent with weaker Eurasian subtropical westerly jet[17–19]. This phenomenon arises from various competing effects, with a major role of the equator-to-polar thermal gradient (also called the meridional temperature gradient), which drives the jet streams through thermal wind balance[1]. A traditional debate concerns the opposing impacts of tropical and Arctic warming on the jet streams, which is referred to as the "tug-of-war" phenomenon[1,20].

Specifically, rapid warming in the Arctic due to sharp sea ice loss diminishes the meridional temperature gradient[16,21,22], while the tropical warming counteracts this due to decreased moist adiabatic lapse rate[23,24]. This tug-of-war has undergone rigorous scientific scrutiny and several studies have shown distinct results[20], such as the insignificant effect of Arctic warming on jet stream and its meanders produced by model simulations[25].

Apart from the aforementioned traditional debate, suppressed convection in tropical Pacific also contributes to the reduced meridional temperature gradient and enhanced waviness of Eurasian jet through a Rossby wavetrain embedded in the jet stream[15]. Moreover, the weakening Eurasian jet is reported to be primarily driven by anthropogenic aerosols[17]. Change in aerosol concentration, particularly decreasing in Europe but increasing in South and East Asia, has led to a decrease in meridional temperature gradient and subsequently a weaker Eurasian jet.

The primary driver of changing Eurasian jet is a multifaceted problem and is still in debate, which emphasizes the need of a deeper insight into the jet stream changes. The regionality of Eurasian jet changes is important[18] due to its expansive range of ~120° in longitude and susceptibility to topography and land-sea distribution, which, however, is generally overlooked. Therefore, the simplistic picture on the whole Eurasian jet changes may obscure notable regional details behind the changes[18,26,27], which impedes our understanding to the changing climate. By focusing on the regionality of changes in the Eurasian jet strength, we discover a sharp increase in interannual variability of the East Asian component (EAJ) of Eurasian jet during high summer, based on seven kinds of statistically significant EAJ mutation data. More importantly, our results highlight that this soaring pulse of EAJ takes the heatwaves, rains and/or draughts across almost the whole Eurasia.

## Results

### Soaring pulse of the East Asia jet

Conceptually, the Eurasian jet has three components: the West Asian sector, East Asian component and western Pacific part (Fig. 1a; see "Methods" and Supplementary Fig. 1 for details). Although the boreal summer is naturally linked to months from June to August, the Eurasian jet axis in July–August apparently shifts further poleward about 5° than that in June (orange vs. green line in Fig. 1a), as well as much weaker jet intensity accompanied by zonal migration of both the Eurasian jet core[28] and the rainy season over the Tibetan plateau[29]. Accordingly, to minimize potential latitudinal bias, we choose the high summer (July–August) period for studying the Eurasian jet.

Apparently, a strong signal emerges over the East Asia (Fig. 1a, b), which is manifested as an interdecadal strongly-increasing interannual variability in EAJ strength (red curve in Fig. 1a), while the West Asian jet and western Pacific jet show a weakening trend (Fig. 1a). These changes in variability are more apparent as we divide the study period into two sub-periods: the pre-1998 period (P1) and the post-1999 period (P2). Specifically, the multi-data mean interannual variability of EAJ has

increased $2.77 \pm 0.10$ m s$^{-1}$ (140% or so; Fig. 1c and Supplementary Table 1), from $2.00 \pm 0.10$ m s$^{-1}$ to $4.77 \pm 0.15$ m s$^{-1}$ (Supplementary Table 2); whereas the WAJ and WPJ do not show significant changes (Fig. 1c). Such change in EAJ interannual variability is robust since all seven reanalysis datasets reach the consensus, with uncertainties of $\pm 0.10$ m s$^{-1}$ among them (Fig. 1c, d and Supplementary Table 1). Interestingly, the three strongest and weakest EAJ years, selected based on the 1.5 standard deviation threshold of the EAJ index (Fig. 3e), all occurs after the late-1990s. The mean strength in the strongest years reaches 34.5 m s$^{-1}$, equivalent to about 173% of that in the weakest years (20.0 m s$^{-1}$). Note that above results are not sensitive to definition region of EAJ since such sharp increase is still apparent when extending its western or eastern boundary (Supplementary Fig. 2). This signal is unique in the context of global subtropical jet stream, since there are no such signals detected in the North Atlantic jet or the Southern Hemispheric jet.

In addition, there is no statistically significant weakening trend in strength over the majority of Eurasian jet ($p > 0.05$ for all three indices in all datasets; Fig. 1c and Supplementary Table 1), inconsistent with the reported weakening summertime Eurasian jet[17,19]. Given that we mainly focus on the change of Eurasian jet in high summer, this result implies that the previously reported weakening summertime Eurasian jet is confined to early summer (June).

Such amplified variability in EAJ reflects a potential change of Eurasian jet. A clear meridional triple structure extends throughout the depth of troposphere (Supplementary Fig. 3a, b), which spans from the East Europe to the East Asia. This can be interpreted as a change in the entire Eurasian jet. Concurrently, strong anomalous easterly wind over the high latitude indicates a weakening polar jet stream, while that over the low latitude indicates an enhancement at the northern flank of tropical upper-level easterly jet.

The upper-level circulation pattern is related to significant changes in heatwave frequency over the Eurasia (Fig. 2a). Especially the majority of Asia shows a large-scale meridional triple structure in both heatwave frequency (Fig. 2a) and surface temperature (Supplementary Fig. 3c). Acting as the background condition for synoptic disturbance, the weakened polar jet (Supplementary Fig. 3a) would impede the propagation of synoptic Rossby waves as well as larger amplitude[22], enhancing synoptic scale wave activity. The consequently enhanced meridional eddy heat mixing[30,31] would lead to warming over the high-latitude. Whereas the intensified subtropical jet (Supplementary Fig. 3a) acts as a stronger "mixing barrier" to inhibit cold (warm) air masses from spilling farther south (intruding poleward)[31]. Consequently, the high-and-low latitudes become warmer, while the mid-latitudes get cooler (Supplementary Fig. 3c), suggesting that the high-and-low latitudes (mid-latitude) of Eurasia may experience an extremely hot summer in years with exceptionally strong (weak) Eurasian jet.

According to the report that the land-atmosphere coupling over the East Asia strengthens since the end of 20st century[32], a heatwave is likely to be accompanied by a drought event. Drought events can rapidly promote heatwave with a 1-day delay, while heatwave often tends to precipitate a drought event with a delay of about 2–7 days[33]. As shown Fig. 2b, the distribution of heatwave anomalies closely aligns with that of the drought mirrored by SPEI, so the abnormal EAJ may be capable of triggering a compound hot and dry extreme event. Thus, the seemingly regional feature of EAJ is related to changes of upper-level circulation over the entire Eurasia, which can inevitably exert impacts on the planetary-scale Eurasian climate.

### Shifting impacts on Eurasian climate

As a key component of East Asian Summer Monsoon[2,34], the EAJ plays a crucial role in climate variability[10,35]. A prevailing notion is that EAJ variability mainly arises from its shifting in latitude, with pulsing in strength as the secondary feature[36]. Nevertheless, pulsing with

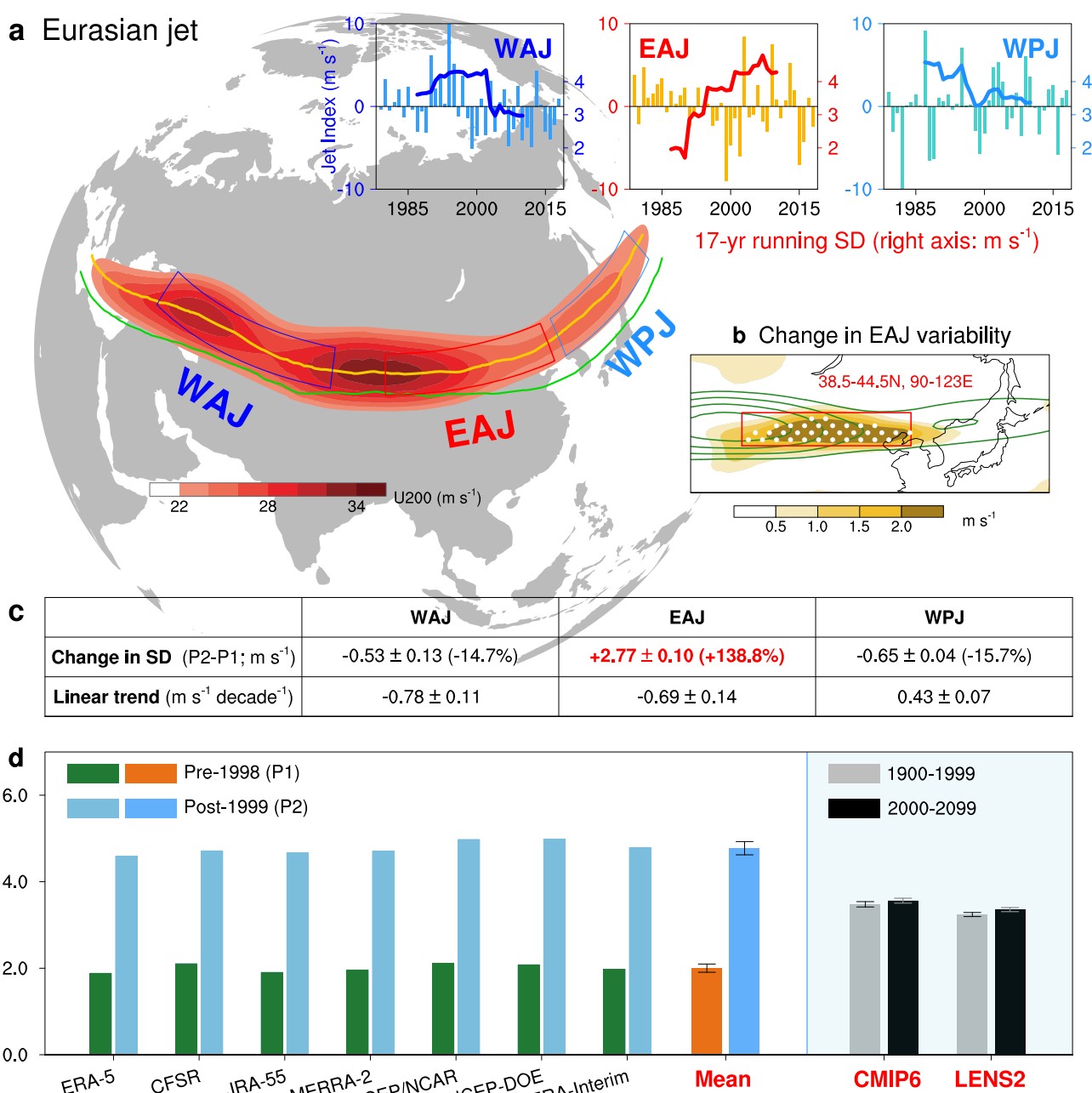

**Fig. 1 | Changes in Eurasian jet variability. a** Climatology of high summer Eurasian jet measured by time averaged 200-hPa zonal wind (U200; shading; units: m s⁻¹) during 1979–2018. Three boxes denote the definition regions of the West Asian jet (WAJ; deep blue box), the East Asian jet (EAJ; red box) and the western Pacific jet (WPJ; light blue box). Eurasian jet axis at 200 hPa in early summer (green line; June) and high summer (yellow line; July–August) are also shown, respectively. Upper row is the year-to-year variation (bars) with 17-year running standard deviation (SD; curves; units: m s⁻¹) in three sub-regions of Eurasian jet. Source data are provided as a Source Data file. **b** Climatology of U200 (contours denote the zonal wind speed from 22 m s⁻¹ to 34 m s⁻¹ by 3 m s⁻¹). Shaded is the epochal difference (P2−P1) in SD of U200, with 95% significance stippled according to the *F*-test. Red box in (**b**) is same as that in (**a**). **c** Changes in variability and linear trend of three jet streams' strength. We use the epochal difference (P2−P1) of SD to measure the change in amplitude, with the percent change of amplitude (difference relative to SD during P1). Bold values denote they are significant at 0.05 confidence level. **d** EAJ variability (measured by SD; units: m s⁻¹) for two sub-periods in multi-source data (see Supplementary Table 2); Error bars of the multi-data mean is the SD (units: m s⁻¹) among seven reanalysis data sets. Results of CMIP6 multi-model mean and LENS2 ensemble mean are also shown, from Supplementary Fig. 13. Source data are provided as a Source Data file.

enhanced amplitude becomes predominant since the late-1990s (Fig. 3a, b). Specifically, dominant spatial form in the pre-1998 epoch indicates an equator-ward shifting EAJ: a classic meridional tripolar structure with alternating signs south and north of the EAJ (Fig. 3a). However, in the post-1999 epoch, this structure shifts northward, with westerly anomaly collocates with the jet axis (Fig. 3b), which is identical to EAJ index in both spatial distribution (Supplementary Fig. 3a) and year-to-year variability (R = 0.98, p < 0.01; Fig. 3e). Similar results

are derived from the EOF analysis on zonal wind anomalies in vertical profile (Supplementary Fig. 4), which suggest that such transitions are not limited at the upper-troposphere, but extend to the whole troposphere.

Transition in the dominant mode of EAJ also leads to the reported shifting rainfall pattern over East Asia, from a meridional triple (Fig. 3c) to a meridional dipole (Fig. 3d) known as the "South Drought-North Flood" pattern[37]. Corresponding to an intensified

**a** Heatwaves

**b** SPEI

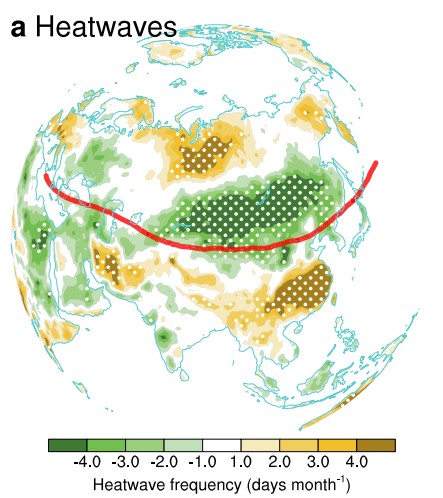

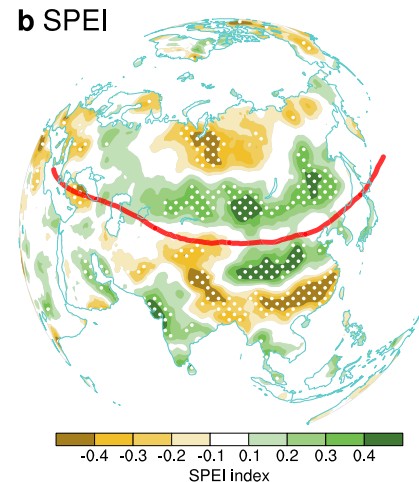

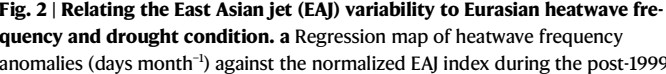

**Fig. 2 | Relating the East Asian jet (EAJ) variability to Eurasian heatwave frequency and drought condition. a** Regression map of heatwave frequency anomalies (days month⁻¹) against the normalized EAJ index during the post-1999 period. **b** Same as (**a**), except for the standardized precipitation evapotranspiration index (SPEI). Red dash line represents the jet axis in 200 hPa in high summer. Dotted areas are statistically significant at the 0.05 level.

EAJ, the East Asia experiences excessive (suppressed) rainfall north (south) of 30°N, as well as excessive rainfall over western Indian subcontinent (Fig. 3d).

This rainfall dipole phenomenon is generated by the abnormal meridional secondary circulation triggered by the pulsing EAJ (Supplementary Fig. 3d). Namely, a stronger EAJ, with strengthened anticyclonic wind shear in its south flank, favors the eastward-extended Tibet Plateau High in the upper-troposphere and the westward-enhanced western North Pacific subtropical high (WNPSH) in the mid-troposphere (Supplementary Fig. 3e), thereby together contributing to strong downdraft in situ (Supplementary Fig. 3d) extending through the depth of the troposphere[38]. Meanwhile, given the enhanced WNPSH, more water vapor is transported to north of ~32°N, leading to more convective precipitation but rainfall deficit to south of ~32°N (Fig. 3d). Besides, the enhanced WNPSH warms subtropical China and the East China Sea via adiabatic subsidence and more incoming solar radiation (Supplementary Fig. 3c). These physical processes establish the heatwave-drought interaction: extreme hot weather intensifies the drought condition while more severe drought exacerbates heatwave through land-atmosphere coupling[32]. Such positive relationship is evident across Eurasia (Figs. 2 and 3d). Given that the land-atmosphere coupling over the East Asia has enhanced since the late-1990s[32], it is expected to see more extreme hot and dry weather in the future.

## Origination from the Scandinavian pattern

The upper-level circulation anomaly over East Asia is often attributed to the upstream North Atlantic sea surface temperature (SST) anomalies[10,26,39,40], via Rossby wavetrain spanning over Eurasia[2,35]. Since the simultaneous correlation does not warrant any causality, the preceding oceanic signal, if existed, can not only help interpreting the causality between ocean and atmospheric, but also provide a source of predictability[40]. As the North Atlantic SST can be induced by large-scale atmospheric teleconnection like the North Atlantic Oscillation[40], the cross-seasonal connection between two atmospheric systems can be established by the oceanic memory effect.

Linearly, we find that abnormal EAJ since the late-1990s is mirrored by another well-known Rossby wave teleconnection in preceding February, called Scandinavian pattern (SCA)[41], which is manifested as a southwest-northeast tilted dipole pattern throughout the troposphere over the North Atlantic (Fig. 4a and Supplementary Fig. 5). As expected, the SCA index is highly correlated with EAJ index ($R = 0.84$, $p < 0.001$; Fig. 4b); and the corresponding pattern

correlation coefficient exceeds 0.98 in each pressure-level over the region 25°N–85°N/65°W–70°E (see Supplementary Table 4 for details). Of note is that, here the strong EAJ variability since the late-1990s is initiated by the SCA in February instead of early spring since the February pattern is distinct and quite different from the typical winter and spring patterns (see Fig.1 in ref. 42).

We estimate the contribution of SCA on the EAJ variability by linearly removing signal of SCA from EAJ (Fig. 4c). As expected, after removal, variability of EAJ declines from $4.77 \pm 0.15\,\mathrm{m\,s^{-1}}$ to $2.53 \pm 0.05\,\mathrm{m\,s^{-1}}$ by $2.25 \pm 0.11\,\mathrm{m\,s^{-1}}$, consistently on seven reanalysis datasets (Supplementary Tables 2 and 3). Since the enhanced variability of EAJ reaches $2.77\,\mathrm{m\,s^{-1}}$ averaged from all datasets (see Fig. 1c and Supplementary Table 1), our primary investigation suggests that the SCA contributes $81.1 \pm 2.9\%$ of the enhanced variability ($2.25\,\mathrm{m\,s^{-1}}$ relative to $2.77\,\mathrm{m\,s^{-1}}$, Supplementary Table 3).

Mechanism involved here is similar to the "cross-seasonal coupled oceanic-atmospheric bridge"[39]. The February SCA mode is mainly confined in the Europe region, with strong high-pressure anomalies centered over the northern Europe (Fig. 4a), which can induce significant SST warming in the northmost Atlantic (Supplementary Fig. 6d). Because the abnormal southeasterly winds along the southwestern flank of the high-pressure anomaly can weaken the climatological southwesterly winds (Fig. 4a and Supplementary Fig. 7a, b) and reduce local sea surface evaporation[42]. Accordingly, the positive phase of SCA can lead to warmer SST via turbulent heat exchange (latent plus sensible heat flux; Supplementary Fig. 7c, d)[42].

Then the so-called memory effect of North Atlantic allows the oceanic signals to persist from spring to summer (Supplementary Fig. 6d–f), and act as oceanic forcing to atmosphere. Although discrepancies exist among datasets, all datasets captured the persistent oceanic signals (Supplementary Fig. 8). The covariance of North Atlantic and pan-Eurasian circulation, obtained by the maximum covariance analysis (see "Methods"), also shows strong SST anomalies around the Iceland and a strengthened EAJ (Supplementary Fig. 9a, b). The connection is established by Rossby wave trains (Supplementary Fig. 9d, e), which originate from North Atlantic and curving down to East Asia, resulting in strong variation of meridional pressure gradient over the major part of Eurasian jet.

Further, we check the physical robustness of this teleconnection by conducting numerical experiments (see "The model and experiment" section in "Methods"). Forced by idealized oceanic warming (Supplementary Fig. 10a) in Northernmost Atlantic, the Atmospheric General Circulation Model simulates a well-organized wavetrain

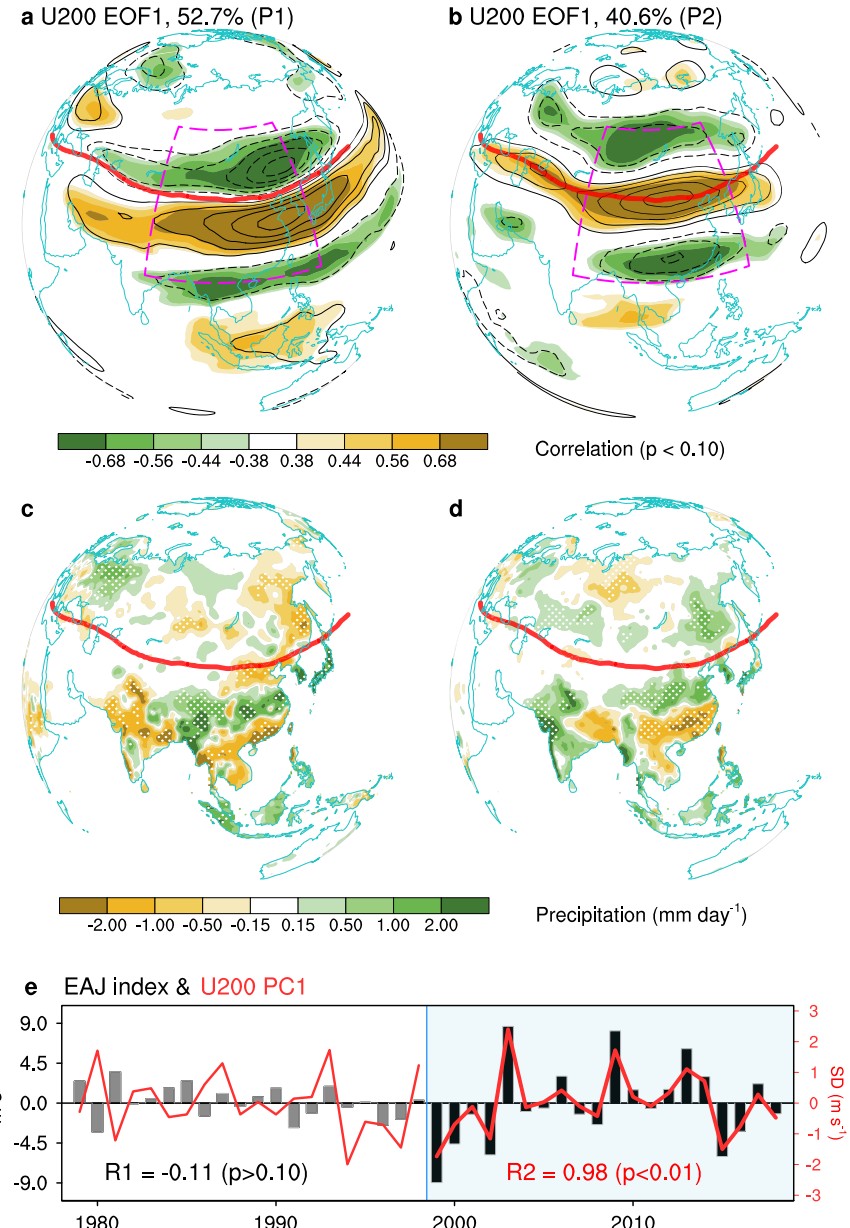

**Fig. 3 | Shifting regimes of large-range Eurasian westerlies and rainfalls.** Spatial distribution of the first empirical orthogonal function (EOF) mode of 200-hPa zonal wind (U200) anomalies (shading: correlation map; contour: regression map) over East Asia for P1 (**a**) and P2 (**b**). Boxes with purple dash line indicates the region for performing EOF decomposition. Spatial distribution of rainfall anomalies related to corresponding first principal component (PC1) for P1 (**c**), and P2 (**d**), with 90% significant region stippled. **e** Interannual time series of detrended East Asian jet index (EAJ; black bar) during 1979–2018. Overlapped red lines represent the corresponding normalized PC1s. Source data are provided as a Source Data file. R1 and R2 is the correlation coefficients between PC1 and EAJ index for P1 and P2, respectively. Red dash line in (**a**–**d**) represents the jet axis in 200 hPa in high summer.

extending from the North Atlantic to East Asia (Supplementary Fig. 10b) and a significantly strengthened EAJ, in spite of some excursions in its position (Supplementary Fig. 10c). Given that the simulated circulation responses over other regions show discrepancies with the observed counterpart (Supplementary Fig. 9), our conclusion on the simulated result should be interpreted carefully.

Such discrepancies can be attributed to the following several aspects. For instance, the mean state in the model exhibits a weaker EAJ but stronger westerlies over higher latitude (Supplementary Fig. 11), which may cause the ray path that originates from the North Atlantic region differs somewhat from the observed one[43,44]. In addition, the Rossby wave response is highly variable with time, which makes it difficult to capture the observed wave structure in model[45].

Finally, the model's resolution, simplifications, assumptions, and parameterizations limit its capability to represent complex climate features[46]. Despite the discrepancies between the simulated wave train pattern and its observed counterpart, this result still strongly suggests that the Northernmost Atlantic around Iceland is indeed a key region generating wavetrain to the East Asia.

To summarize, this mechanism is similar to the concept "cross-seasonal coupled oceanic-atmospheric bridge" (Supplementary Fig. 12): the North Atlantic acts as an oceanic bridge that stores the preceding atmospheric signal (the Scandinavian pattern), and releases the signal in the following season via complex oceanic dynamic and thermal process. This results in relevant atmospheric teleconnection modulating the EAJ[39,40].

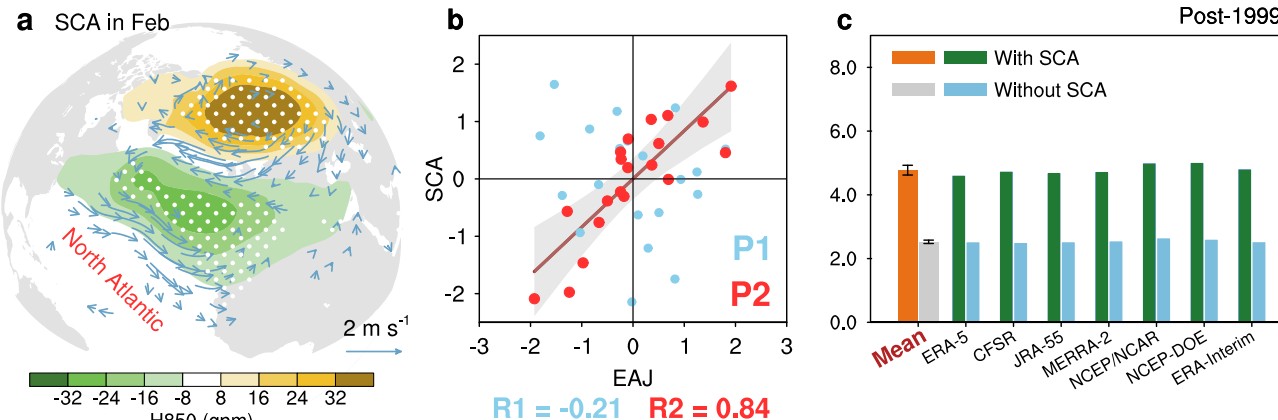

**Fig. 4 | Contribution of North Atlantic seesaw pattern (SCA) to the enhanced East Asian jet (EAJ) variability. a** Regression maps of February 850-hPa geopotential height (H850; units: gpm) and surface wind (only show vectors with $p < 0.10$; units: m s$^{-1}$) anomalies against the EAJ index during P2. Dotted areas are statistically significant at the 0.05 confidence level. **b** Scatter diagram for normalized EAJ and SCA indices during P1 (light blue dots) and P2 (red dots). The solid line is an ordinary least-squares fit for P2, with shading as the 95% confidence interval. Source data are provided as a Source Data file. **c** EAJ variability, measured by standard deviation (SD; units: m s$^{-1}$) for the post-1999 period, with the original variability (green bar) and that after linearly removing SCA signal (light blue bar), see Supplementary Table 2. Error bars of the multi-data mean is the SD among datasets.

## Discussion

Here we identified an unprecedented surge in the variability of EAJ intensity since the late-1990s (by about 140%), which is mainly associated with the Scandinavian pattern in preceding February, via the "cross-seasonal coupled oceanic-atmospheric bridge". The emergence of "coupling" between SCA and EAJ may arise from the remarkable non-stationary relationship between the SCA and North Atlantic SST (Supplementary Fig. 6). Although in both periods the SCA in preceding February can induce oceanic anomaly persisting to ensuing summer, it failed to establish the "bridge" prior to the late-1990s due to absence of robust SST anomalies at the key region of exciting Rossby wave train[47] (red box in Supplementary Fig. 6c, f).

Whether such phenomenon can be well captured by climate models is of great importance for proper future projection. Here we conduct a preliminary investigation on the future change of EAJ variability under global warming, utilizing 50 models of the Coupled Model Intercomparison Project Phase 6 (CMIP6; Supplementary Table 5; see "Methods"). These state-of-the-art CMIP6 models do not reach the consensus on how the variability of EAJ intensity will change in future high-emission scenarios (Fig. 1d and Supplementary Fig. 13a). To rule out the influence of model uncertainty and focus on the internal variability, we also use the CESM2-Large Ensemble with 50 members. Although those members show a strengthening of variability during the SSP370 scenarios, barely significant at 0.05 confidence level (Fig. 1d and Supplementary Fig. 13b), the amplitude of strengthening is much smaller (4.1% relative to observation). Thus, although the EAJ's variability may increase under global warming, the majority of change in variability is more likely to arise from unforced natural internal process instead of external forcing.

## Methods
### Data
This study is aimed at exploring the interannual variability of summer EAJ and relevant potential mechanisms. Monthly and/or daily atmospheric data used include the European Centre for Medium-Range Weather Forecasts' (ECMWF's) fifth-generation reanalysis (ERA-5[48]) at a high resolution of 0.25° × 0.25° in this study, the Japanese 55-year reanalysis (JRA-55[49]) with longitude-latitude resolution of 1.25° × 1.25°, the National Centers for Environmental Prediction (NCEP) Climate Forecast System Reanalysis (CFSR[50,51]) with the spatial resolution of 0.5° × 0.5°, the National Centers for Environmental Prediction-National Center for Atmospheric Research (NCEP-NCAR[52]) Reanalysis 1 data with the spatial resolution of 2.5° × 2.5°, the National Centers for Environmental Prediction/Department of Energy (NCEP/DOE[53]) Reanalysis 2 data with the spatial resolution of 2.5° × 2.5°, the Modern-Era Retrospective Analysis for Research and Applications version 2 (MERRA2[54]) with the spatial resolution of 0.625° × 1° and the ERA-interim Reanalysis[55] with the spatial resolution of 1° × 1°.

The monthly mean precipitation data employed in this study are from the National Oceanic and Atmospheric Administration (NOAA)'s Precipitation Reconstruction over Land (PREC/L) on a 1.0° × 1.0° grid[56] and the Global Precipitation Climatology Program (GPCP[57]) with a 2.5° × 2.5° resolution.

Sea surface temperature (SST) datasets used includes the NOAA Extended Reconstructed SST version 5 (ERSST.v5[58]; 2° × 2°), the Hadley Centre SST (HadISST[59]; 2.5° × 2.5°), OISST.v2[60] (1° × 1°) and COBE-SST2[61](1° × 1°).

We use the standardized precipitation evapotranspiration index (SPEI) to measure the drought conditions over Eurasia, which are obtained from the Global SPEI database[62] of 1 month timescale at resolution of 0.25° × 0.25°. The SPEI was computed based on monthly precipitation and temperature data obtained from the CRU TS4.04 data, taking the linkage between the temperature and the drought severity into consideration.

### Indices
The WAJ, EAJ and WPJ index is defined as the regional average of U200 anomalies over the specific region (see the boxes in Fig. 1a) in high summer during 1979–2018. Of note is that the region chosen for defining EAJ index is located at the downstream of climatological EAJ maximum, i.e., the so-called exit region of EAJ[63], not cover the whole EAJ maximum. The selection criteria are mainly based on the following two points: (1) The standard deviation (SD) of zonal wind at 200 hPa (U200) is relative weak over the EAJ maximum region (SD less than 3.5 m s$^{-1}$, Supplementary Fig. 1), suggesting a strong and stable jet stream in this region; (2) Moreover, there is also no significant inter-decadal change in SD of U200 over the climatological EAJ maximum region (Fig. 1b). Accordingly, the EAJ index is defined as the regional average of U200 anomalies over the exit region of EAJ (38.5–44.5°N, 90–123°E), with robust interdecadal changes in SD (Fig. 1b).

The North Atlantic Oscillation (NAO) index was downloaded from https://psl.noaa.gov/data/correlation/nao.data. The Scandinavian (SCA) index was downloaded from https://www.cpc.ncep.noaa.

gov/data/teledoc/scand.shtml. The AMO index was downloaded from https://climatedataguide.ucar.edu/climate-data/atlantic-multi-decadal-oscillation-amo.

### EOF, MCA, correlation, regression and composite analysis

The empirical orthogonal function (EOF) analysis is performed to extract the leading modes of EAJ. It is used to obtain the leading modes of U200 anomalies over East Asia (20°–60°N, 80°–130°E), and the leading modes of zonal wind anomalies in meridional section (15°–65°N, 1000-hPa-100-hPa; averaged from 80°E to 130°E).

The maximum covariance analysis (MCA), also called the singularly valuable decomposition[64,65], is used to obtain the dominant coupled modes between North Atlantic SST (35°–85°N, 50°W–14°E) and U200 (10°–80°N, 10°W–140°E) over the Eurasian continent during high summer.

In this study, the statistical significance of linear regression and Pearson correlation is evaluated by two-tailed Student's $t$ test. The degrees of freedom for two sub-periods are both 18; thus, the correlation coefficients corresponding to 90%, 95%, 99% and 99.9% confidence level are 0.38, 0.44, 0.56 and 0.68, respectively. As for the difference of the standard deviation (SD), the Fisher's $F$-test is applied for analyzing its significance.

For composite analysis (Supplementary Fig. 3e), years for strong (weak) EAJ are defined as the year whose corresponding values of normalized EAJ index is above +1.5 (below −1.5). There are both three years for intensified EAJ (2003, 2009, 2013) and the weakened (1999, 2002, 2015).

### Heatwave frequency

The heatwave frequency at one grid is defined as the number of days whose maximum temperature is above the threshold (90th percentile of the records of that calendar day at 5 days window[32]). It was calculated based on the ERA-5 daily reanalysis data over the period 1979–2018.

### Rossby wave activity flux

To describe the energy propagation of the quasi-stationary Rossby waves, the horizontal component of the wave activity flux (WAF[66,67]) is calculated with monthly mean data in this study. The horizontal component of the WAF in pressure coordinates is expressed as:

$$WAF = \frac{p}{2000|\vec{U}|} \begin{cases} U\left(\psi'^2_x - \psi'\psi'_{xx}\right) + V(\psi'_x\psi'_y - \psi'\psi'_{xy}) \\ U\left(\psi'_x\psi'_y - \psi'\psi'_{xy}\right) + V\left(\psi'^2_y - \psi'\psi'_{yy}\right) \end{cases} \quad (1)$$

where $\psi'$ represents the stream function of quasi-geostrophic flow.

### The models and experiment

The numerical experiment is performed to investigate the atmospheric response to anomalous SST warming, using the Community Atmosphere Model version 4 (CAM4[46]), the atmospheric component of the Community Earth System Model version 1.2.2 (CESM1.2.2) from the National Center for Atmospheric Research (NCAR). The experiments utilized a simulation framework based on the F_2000 component set, which incorporated 26 vertical sigma levels with a horizontal resolution of 1.9° × 2.5°. These simulations were driven by prescribed region-specific SST anomalies, without feedbacks from atmosphere to the ocean.

A 30-year control run (CTRL) is constituted using observed climatological annual cycled monthly mean SST to obtain the model's atmospheric climatology. The sensitivity run (EXP) is the same as the CTRL, but forced by the climatological SST plus prescribed SST anomalies around the Iceland (Supplementary Fig. 10a) in July–August. The SST anomalies is considered as an idealized forcing in July–August, using a box covering the region (56°–68°E, 26°W–0). The forcing magnitude is obtained as the regression coefficient of SST anomalies against the normalized EAJ index, and then scaled by a factor of 2. The

EXP was run for 30 years with outputs of the last 25 years used for analysis. The impact of the given anomalous SST forcing is identified by differences between the EXP and CTRL (Supplementary Fig. 10b, c). The statistical significance of results is evaluated by two-tailed Student's $t$ test.

The Coupled Model Intercomparison Project phase 6 (CMIP6[68]) is used to exam its ability in simulating the EAJ in present climate and future change. It simulates the historical climate from 1850 to 2014 and future climate from 2015 after under different scenarios. We select the present period (1900–1999) and the future period (2000–2099) in SSP585 (Shared Socioeconomic Pathways 5-8.5) scenario for the investigation on EAJ variability under the influence of anthropogenic warming, based on 50 models (Supplementary Table 5). For each model, we firstly identify its EAJ axis using climatological U200 during 1979–2014 and obtain the latitude of EAJ. Therefore, the definition region of EAJ in each model is determined by the box centered at the corresponding jet latitude with a 10° range in latitude and covering the longitudinal range 90°–123°E. Focusing on the interannual variability, the trend and decadal variability are removed before calculating the SD. Changes of SD between present and future periods is considered as change in EAJ interannual variability under anthropogenic warming. The significance test is conducted using the bootstrap method.

In addition, the CESM2 Large Ensemble Community Project (LENS2) is used to focus the internal variability, using 50 members at 1-degree spatial resolution. We also select the present period (1900–1999) and the future period (2000–2099) in SSP370 (Shared Socioeconomic Pathways 3-7.0) scenario.

### Bootstrap test

The bootstrap method[69] is conducted to examine whether the changes in EAJ strength and variation are statistically significant in CMIP6 models and LENS2 members. Results from 50 models were resampled randomly and averaged to get a total of 10,000 realizations. Any model is allowed to be selected again. The SD of the 10,000 realizations for each period is computed respectively. If the difference of multi-model mean between two sub-periods is larger than sum of the two separate SD values, then such change is statistically significant at 0.05 confidence level.

### NCAR Command Language (NCL)

NCL[70] is a free and open-source data analysis and visualization software developed by the NCAR. It provides a wide range of functionalities for scientific data analysis, including file input and output, data processing, statistical analysis, and data visualization. It supports a broad range of data formats, including netCDF, HDF, GRIB, and others. It provides a large number of built-in functions and procedures, which allows users to write scripts to perform complex tasks with relatively few lines of code. It also offers powerful visualization capabilities. It includes a wide range of graphical techniques for data presentation, such as contour plots, color-shaded plots, vector plots, and others. The graphics are highly customizable, allowing users to control virtually every aspect of the plot. Introduction, installation, usage and demo of using NCL are provided in Supplementary Code 1.

All base maps in this study are draw directly from the NCL.

## Data availability

The ERA-5 data are available at https://rda.ucar.edu/datasets/ds628.0/. The ERA-Interim data are available at https://www.ecmwf.int/en/forecasts/dataset/ecmwf-reanalysis-interim. The JRA-55 data are available at https://cds.climate.copernicus.eu/. The MERRA-2 data are available at https://disc.gsfc.nasa.gov/datasets?project=MERRA-2&descriptionFromFileType=Attached%20File. The NCEP-NCAR Reanalysis 1 data are available at https://psl.noaa.gov/data/gridded/data.ncep.reanalysis.html. The NCEP/DOE Reanalysis 2 data are available at https://psl.noaa.gov/data/gridded/data.ncep.reanalysis2.html. The

CFSR data are available at https://rda.ucar.edu/datasets/ds093.2/. The CFSV2 data are available at https://rda.ucar.edu/datasets/ds094.2/. The monthly NOAA's Precipitation Reconstruction over Land (PREC/L) is available at https://psl.noaa.gov/data/gridded/data.precl.html. The monthly GPCP Version 2.3 Combined Precipitation data are available at https://psl.noaa.gov/data/gridded/data.gpcp.html. The NOAA ERSST V5 are available at https://www.esrl.noaa.gov/psd/data/gridded/data.noaa.ersst.v5.html. The Hadley Centre Sea Ice and Sea Surface Temperature data set are available at https://www.metoffice.gov.uk/hadobs/hadisst/. The COBE SST2 are available at https://psl.noaa.gov/data/gridded/data.cobe2.html. The NOAA OI SST V2 are available at https://psl.noaa.gov/data/gridded/data.noaa.oisst.v2.highres.html. The simulation outputs of CMIP6 are available at https://esgf-node.llnl.gov/search/cmip6/. The simulation outputs of LENS2 are available at https://www.cesm.ucar.edu/projects/community-projects/LENS2/ The Global SPEI database is are available at https://digital.csic.es/handle/10261/332007. Source data are provided with this paper.

## Code availability

The CESM model can be downloaded from http://github.com/ESCOMP/CESM. The NCAR Command Language (Version 6.6.2) is used for plotting. Codes to reproduce these figures are available at https://zenodo.org/doi/10.5281/zenodo.10552597.

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

## Acknowledgements

This work is supported by the National Natural Science Foundation of China (Grant Nos. 41975077 and 42088101), the Innovation Group Project of Southern Marine Science and Engineering Guangdong Laboratory (Zhuhai) (Grant No. 311022001), Guangdong Province Key Laboratory for Climate Change and Natural Disaster Studies (Grant No.2020B1212060025) and Open Foundation of State Key Laboratory of Satellite Ocean Environment Dynamics, Second Institute of Oceanography, MNR (Grant No. QNHX2331). We are grateful to the research start-up funding support of the "Top 100 Talents Plan" Project of Zhejiang University and the high-performance computing condition of the Ocean College at Zhejiang University, as well as the support from Jiangsu Collaborative Innovation Center for Climate Change.

## Author contributions

C.H. conceived and designed the study. L.L. and C.H. conducted the analysis and drafted the manuscript. B.W., R.W., S.Y., W.C. and D.C. provided comments and revised the manuscript. B.W., R.W. and W.C. edited the paper. L.L., C.H., B.W., R.W., Z.W., S.Y., W.C., P.L., X.X. and D.C. discussed the scientific interpretation of the results.

## Competing interests

The authors declare no competing interests.
