## [Peer Review File · Nature Communications]

Atlantic origin of the increasing Asian westerly jet interannual variabilityEditorial Note: Parts of this Peer Review File have been redacted as indicated to remove third-party material where no permission to publish could be obtained.

REVIEWER COMMENTS

Reviewer #1 (Remarks to the Author):

Review's comments for the revised paper (NCOMMS-23-57943-T), entitled "Atlantic origin of the sharply increasing Asian westerly jet variability", submitted to Nature Communications

By using observations/reanalyses, climate model simulations and a pair of atmospheric model simulations, this paper investigated decadal changes of interannual variability of the East Asian jet (EAJ) in summer (July-August). The results based on various reanalysis data sets indicated a robust and consistent enhanced EAJ interannual variability and this variability is associated with different patterns of heat waves and drought conditions over the Eurasia continent. Further, the mechanisms that are responsible for increased variability were elucidated based on both reanalyses and atmospheric sensitivity experiments. However, results from climate model simulations do not show this increased variability and the paper concluded that the recent increasing in the East Asian jet variability is mainly attributed to the unforced natural variability. The results are interesting and important for climate science community. The paper is worth for publication. However, the paper needs some improvements by properly addressing the comments listed below before it can be accepted for publication in Nature Communications.

Major comments

1. The study is about interannual variability. The time scale needs to be explicit in the title, abstract, and results, and methods.
2. Figure 1a. The region used to define EAJ index only covers part of EAJ and is different from the traditional region which covers whole EAS jet maximum. Figure 1b shows that increased variability is in downstream of the climatological EAJ maximum. Therefore, conclusions drawn from the study must be sensitive how to define the EAJ index. Authors need to make this clear and make justification to avoid misunderstanding.
3. Some analyses are based on regressions, correlations, and composites simultaneously. The causality is not clear and therefore conclusions based on these shall be drawn very carefully.
4. Some of key statements on changes associated with increased summertime EAJ interannual variability need to be quantified.
5. Linear trends of WAJ in figure 1c and those listed in the Extended Table 1 are not consistent, with the magnitude in figure 1c being about 10% of individual data sets displayed in the Extended Table 1. Therefore, conclusions based on figure 1c are not correct. Please check and make corrections.

Specific comments

1. Line 30. "(contributing to $81.1\% \pm 2.9\%$ of the enhanced EAJ variability". See specific comment 6.
2. Lines 33-35. It shall be about increasing East Asian jet variability not the jet itself. Rephrase.
3. "high summer" and "peak summer" are used in text to refer July-August. Better to use one.
4. Lines 97-101. See major comment 5.

5. Line 106-107. Tropical easterly jet only occurs in the upper troposphere. Be more specific.
6. Lines 161. The reviewer could not understand how authors get that the SCA contributes 81.1% of the enhanced variability (2.25 m s⁻¹ relative to 2.77 m s⁻¹). Where does 2.77 ms⁻¹ come from?
7. Lines 180-181. The zonal wind responses over other regions (Extended Data Fig. 6c) are different from those based on reanalyses (Extended Data Fig. 5b). Need to make some comments on this aspect.
8. Line 337-338. Why do authors use two different longitude ranges?
9. Fig. 1c. Linear trend for WAJ is -0.08 m s⁻¹ per decade. See major comment 5 and specific comment 15.
10. Fig. 2. Are these regressions or composites of heat wave frequency and drought conditions? Clarify.
11. Fig. 3. Use the same latitude range in panel b and d.
12. Line 558. "natural state TPH", what is it? Do you mean neutral year of EAJ or climatology?
13. Extended Data Fig. 5. Please plot climatological jet axis in panel b. The sub-label "Geopotential Height" in panel b is not correct.
14. Extended Data Fig. 6. Please plot model climatological jet axis in panel c.
15. Extended Data Table 1. The WAJ trends range from -0.65 to -0.98 m s⁻¹ per decade. See major comment 5 and specific comment 9.

Typos

1. Line 370. "Ice Land" to "Iceland"
2. Line 602, "Ice Land" to "Iceland"

Reviewer #2 (Remarks to the Author):

This study reported an unprecedented strong variability of EAJ intensity since the late-1990s, which is closely associated with large-range climate extremes in East and South Asia, and its driving factor of the northernmost Atlantic SSTA. This study also revealed that the persistent SSTA signals of EAJ variability the late-1990s are initiated by the Scandinavian pattern in February. The authors have demonstrated in detail the intrinsic correlation between the Scandinavian pattern in February, the persistent North Atlantic SSTA anomalies and the EAJ variability in high summer, under the hypothesis of "cross seasonal coupled oceanic atmospheric bridge". The argumentation and the conclusions are convincing. This study tells a complete and interesting story, which has important reference value. It should be considered for publication after the following revisions.

Major concerns

- (1) Although I am convinced that the Northernmost Atlantic around Iceland is indeed a key region of the SSTA forcing of generating the strong EAJ variability, the simulated wave train response somehow differs from its observed counterpart. How to justify it?
- (2) The strong EAJ variability since the late-1990s was initiated by the Scandinavian pattern in February instead of early spring. The February pattern is distinct and quite different from the typical winter and spring patterns. The authors should be with caution.

Minor concerns

L123-124: which is identical to EAJ in both spatial distribution (Extended Data Fig. 3a)—the figure legend is not correct.

L131-134: “A stronger EAJ, with strengthened anticyclonic wind shear in its south flank, can induce the eastward-extended Tibet Plateau High in the upper-troposphere, enhanced western North Pacific subtropical high in the mid-troposphere and significant warming near surface (Extended Data Fig. 1c, e)” --- the authors should be careful whether there is a causal relationship among them.

L156-157: “with almost identical spatial pattern (Extended Data Fig. 2a; pattern correlation coefficient reaches 0.99)” --- it needs to be clarified.

Response to Reviewers' Comments

Response to reviewer #1:

Review's comments for the revised paper (NCOMMS-23-57943-T), entitled "Atlantic origin of the sharply increasing Asian westerly jet variability", submitted to Nature Communications

By using observations/reanalyses, climate model simulations and a pair of atmospheric model simulations, this paper investigated decadal changes of interannual variability of the East Asian jet (EAJ) in summer (July-August). The results based on various reanalysis data sets indicated a robust and consistent enhanced EAJ interannual variability and this variability is associated with different patterns of heat waves and drought conditions over the Eurasia continent. Further, the mechanisms that are responsible for increased variability were elucidated based on both reanalyses and atmospheric sensitivity experiments. However, results from climate model simulations do not show this increased variability and the paper concluded that the recent increasing in the East Asian jet variability is mainly attributed to the unforced natural variability. The results are interesting and important for climate science community. The paper is worth for publication. However, the paper needs some improvements by properly addressing the comments listed below before it can be accepted for publication in Nature Communications.

Response: Thank you for your positive and supportive comments and detailed instruction on how to improve the manuscript. The quality of the manuscript has been greatly improved based on your comments. Please find our response below.

Major comments

1. The study is about interannual variability. The time scale needs to be explicit in the title, abstract, and results, and methods.

Response: Thank you for pointing out this issue. This study indeed focuses on changes in the interannual variability of EAJ. According to your suggestion, we have thoroughly revised the manuscript to clarify this aspect. Nonetheless, to avoid the title being too long, we did not include more information in the title, such as the time scale and specific season. For example, the following statement has been added into the Abstract, Results and the Methods:

Lines 32-34: “This sheds light on that, at the interannual timescale, a substantial portion of recently increasing variability in the East Asian sector of the Eurasian westerly jet arises from unforced natural variability.”

Lines 93-98: “Specifically, the multi-data mean interannual variability of EAJ has increased $2.77 \pm 0.10 \text{ m s}^{-1}$ (about 139%; **Fig. 1c** and **Supplementary Table 1**), from $2.00 \pm 0.10 \text{ m s}^{-1}$ to $4.77 \pm 0.15 \text{ m s}^{-1}$ (**Supplementary Table 2**); whereas the WAJ and WPJ do not show significant changes (**Fig. 1c**). Such change in EAJ interannual variability is robust since all seven reanalysis datasets reach the consensus, with uncertainties of $\pm 0.10 \text{ m s}^{-1}$ among them (**Fig. 1c, d** and **Supplementary Table 1**).”

Lines 258-259: “This study is aimed at exploring the interannual variability of EAJ and relevant potential mechanisms.”

2. Figure 1a. The region used to define EAJ index only covers part of EAJ and is different from the traditional region which covers whole EAS jet maximum. Figure 1b shows that increased variability is in downstream of the climatological EAJ maximum. Therefore, conclusions drawn from the study must be sensitive how to define the EAJ index. Authors need to make this clear and make justification to avoid misunderstanding.

Response: Thank you for this helpful comment. As shown in **Fig. R1** (i.e., **Supplementary Fig. 1**), the variability is very weak over the EAJ maximum region (SD less than 3.5 m s^{-1}), suggesting a strong and stable jet stream in this maximum

region. Moreover, there is no significant interdecadal change in SD of U200 over the climatological EAJ maximum region (**Fig. 1b**). In other words, even if the EAJ maximum region were included in to the definition region of EAJ index, the observed sharply increasing interannual variability of EAJ would not be sensitive how to define the EAJ index. Nevertheless, to test whether our conclusions are sensitive to the choice of definition region. We further show the EAJ index using two different longitudinal ranges. As shown in **Fig. R2 (i.e., Supplementary Fig. 2)**, It is clear that the sharply increase in interannual variability of EAJ also keeps robust when its definition region is selected as 85–123°E (**Fig. R2a**) or even 80°–123°E (**Fig. R2b**), suggesting that our conclusions are indeed not sensitive to the definition of EAJ index. And this additional analysis further reinforces the robustness of our results. We have added relevant explanations to the revised manuscript as follows:

Lines 284-293: “Of note is that the region chosen for defining EAJ index is located at the downstream of climatological EAJ maximum, i.e., the so-called exit region of EAJ²⁸, not cover the whole EAJ maximum. The selection criteria are mainly based on the following two points: (i) The standard deviation (SD) of zonal wind at 200 hPa (U200) is relative weak over the EAJ maximum region (SD less than 3.5 m s⁻¹, **Supplementary Fig. 1**), suggesting a strong and stable jet stream in this region; (ii) Moreover, there is also no significant interdecadal change in SD of U200 over the climatological EAJ maximum region (**Fig. 1b**). Accordingly, the EAJ index is defined as the regional average of U200 anomalies over the exit region of EAJ (38.5–44.5°N, 90–123°E), with robust interdecadal changes in SD (**Fig. 1b**).”

Fig. R1 | Jet stream variability measured by SD in U200 (unit: ms^{-1}). Three boxes denote definition regions of WAJ, EAJ and WPJ, same as in Fig.1a.

Fig. R2 | Sensitivity of changes in inter-annual variability of EAJ index due to definition region. Shown in each sub-figure is the year-to-year variations (yellow bars) and the corresponding 17-yr running SD (red curves; units: m s^{-1}) of EAJ index. Definition regions are shown in titles of each sub-figure.

Reference:

28. Wang, W., Zhou, W., Wang, X., Fong, S. K. & Leong, K. C. Summer high temperature extremes in Southeast China associated with the East Asian jet stream and circumglobal teleconnection. *J. Geophys. Res.: Atmos.* **118**, 8306–8319 (2013).

3. Some analyses are based on regressions, correlations, and composites simultaneously. The causality is not clear and therefore conclusions based on these shall be drawn very carefully.

Response: Thank you for pointing out this issue. We agree that the simultaneous analyses do not guarantee causality. Thus we have rewritten relevant statements in the revised manuscript as follows:

Lines 117-127: “The upper-level circulation pattern is related to significant changes in heatwave frequency over the Eurasia (**Fig. 2a**). Especially the majority of Asia shows a large-scale meridional triple structure in both heatwave frequency (**Fig. 2a**) and surface temperature (**Supplementary Fig. 3c**). Since the weakened polar jet (**Supplementary Fig. 3a**) favors more synoptic disturbances²², larger amplitude planetary waves and meridional eddy mixing³¹, which leads to warming over the high-latitude; whereas the intensified subtropical jet (**Supplementary Fig. 3a**) acts as a stronger “mixing barrier” to inhibit cold (warm) air masses from spilling farther south (intruding poleward)³². Consequently, the high-and-low latitudes become warmer, while the mid-latitudes get cooler (**Supplementary Fig. 3c**), suggesting that the high-and-low latitudes (mid-latitude) of Eurasia may experience an extremely hot summer in years with exceptionally strong (weak) Eurasian jet.”

Lines 128-133: “According to the report that the land-atmosphere coupling over the East Asia strengthens since the end of 20st century³³, a heatwave is likely to be accompanied by a drought event. Drought events can rapidly promote heatwave with a 1-day delay, while heatwave often tends to precipitate a drought event with a delay of about 2–7 days³⁴. As shown **Fig. 2b**, the distribution of heatwave anomalies closely aligns with that of the drought mirrored by SPEI, so the abnormal EAJ may be capable of triggering a compound hot and dry extreme event.”

Lines 153-162: “This rainfall dipole phenomenon is generated by the abnormal meridional secondary circulation triggered by the pulsing EAJ (**Supplementary**

Fig. 3d). Namely, a stronger EAJ, with strengthened anticyclonic wind shear in its south flank, favors the eastward-extended Tibet Plateau High in the upper-troposphere and the westward-enhanced western North Pacific subtropical high (WNPSH) in the mid-troposphere (**Supplementary Fig. 3e**), thereby together contributing to strong downdraft in-situ (**Supplementary Fig. 3d**) extending through the depth of the troposphere³⁹. Meanwhile, given the enhanced WNPSH, more water vapor is transported to north of $\sim 32^\circ\text{N}$, leading to more convective precipitation there but rainfall deficit to south of $\sim 32^\circ\text{N}$ (**Fig. 3d**). Besides, the enhanced WNPSH warms subtropical China and the East China Sea via adiabatic subsidence and more incoming solar radiation (**Supplementary Fig. 3c**).”

4. Some of key statements on changes associated with increased summertime EAJ interannual variability need to be quantified.

Response: Done as suggestion. First, we have calculated and added the interannual variability of EAJ in different periods using all datasets, as shown **Table R1 (i.e., Supplementary Table 2)**, and rewritten the sentences on the increased summertime EAJ interannual variability as follows:

Line 91-96: “These changes in variability are more apparent as we divide the study period into two sub-periods: the pre-1998 period (P1) and the post-1999 period (P2). Specifically, the multi-data mean interannual variability of EAJ has increased $2.77 \pm 0.10 \text{ m s}^{-1}$ (about 139%; **Fig. 1c** and **Supplementary Table 1**), from $2.00 \pm 0.10 \text{ m s}^{-1}$ to $4.77 \pm 0.15 \text{ m s}^{-1}$ (**Supplementary Table 2**); whereas the WAJ and WPJ do not show significant changes (**Fig. 1c**).”

Table R1 (i.e., Supplementary Table 2) | Change in EAJ inter-annual variability in different datasets. The EAJ index used are described in “Indices” section of Methods. The inter-annual variability is determined by the SD (unit: m

s-1). The mean values are shown in red and bold.

Datasets	P1	P2	P2 (SCA removed)
ERA-5	1.88	4.59	2.50
JRA-55	1.90	4.67	2.47
ERA-I	1.98	4.79	2.50
NCEP-NCAR	2.11	4.98	2.52
NCEP/DOE	2.08	4.99	2.62
MERRA-2	1.96	4.70	2.58
CFSR	2.10	4.71	2.50
Mean	2.00 ± 0.10	4.77 ± 0.15	2.53 ± 0.05

5. Linear trends of WAJ in figure 1c and those listed in the Extended Table 1 are not consistent, with the magnitude in figure 1c being about 10% of individual data sets displayed in the Extended Table 1. Therefore, conclusions based on figure 1c are not correct. Please check and make corrections.

Response: Thank you for pointing out this issue. Then we further checked our code and found that we made a mistake in calculating the changes in WAJ and WPJ in the old version of Fig. 1c. Now we have made corrections correspondingly. In fact, the magnitude of WAJ in the old version Fig. 1c actually represents the 0.078 (Rounded to 0.08) $m s^{-1} year^{-1}$, it has been corrected to $0.78 m s^{-1} decade^{-1}$ to match the unit. In addition, we have re-checked all the results in Fig.1c and Supplementary Table 1 (i.e., the old Extended Table 1), and consequently corrected all errors, including changes in SD for WAJ and WPJ in all reanalysis data.

Nevertheless, it remains that only EAJ exhibits a strong increase in its inter-annual variability and none of them exhibits a significant linear trend. Besides, we have also double-checked all analysis codes in this work and are confident that all the other results have been correctly processed.

Specific comments

1. Line 30. “(contributing to $81.1\% \pm 2.9\%$ of the enhanced EAJ variability”. See specific comment 6.

Response: Please see the response to the following specific comment 6.

2. Lines 33-35. It shall be about increasing East Asian jet variability not the jet itself. Rephrase.

Response: Agree. We have made modifications on this sentence as follows:

Lines 32-34: “..., This sheds light on that, at the interannual timescale, a substantial portion of recently increasing variability in the East Asian sector of the Eurasian westerly jet arises from unforced natural variability.”

3. “high summer” and “peak summer” are used in text to refer July-August. Better to use one.

Response: Done as suggestion. We have replaced all "peak summer" by "high summer" to ensure consistency in terminology.

4. Lines 97-101. See major comment 5.

Response: Corrected. Please see the specific response to above major comment 5.

5. Line 106-107. Tropical easterly jet only occurs in the upper troposphere. Be more specific.

Response: We agree that the tropical easterly jet is confined over the upper-troposphere. As shown in following **Fig. R3**, the easterly jet is indeed located at approximate 10°N . Therefore, the easterly wind anomalies should be interpreted as enhancement at the northern flank of tropical easterly jet. Then we have rewritten this sentence as follows:

Lines 114-116: “Concurrently, strong anomalous easterly wind over the high latitude indicates a weakening polar jet stream, while that over the low latitude indicates an enhancement at the northern flank of tropical upper-level easterly jet.”

Fig. R3 | Climatology of U200 (m s⁻¹) during high summer.

6. Lines 161. The reviewer could not understand how authors get that the SCA contributes 81.1% of the enhanced variability (2.25 m s⁻¹ relative to 2.77 m s⁻¹). Where does 2.77 ms⁻¹ come from?

Response: We apologize for the previously ambiguous explanation on the conclusion drawn on SCA's contribution to the enhanced variability of EAJ. Initially, we found that interannual variability of EAJ has increased from 2.00 ± 0.10 m s⁻¹ to 4.77 ± 0.15 m s⁻¹ by **2.77** m s⁻¹ (**Table R1**; also shown in **Fig.1c**)

However, after removing the SCA signal from EAJ, the variability reduced by **2.25** \pm 0.11 m s⁻¹ (see **Table R2**), a value attributed to the contribution of SCA to EAJ variability. Consequently, the averaged contribution ratio is calculated as **2.25/2.77** \approx **81.1%** (**Table R2**). We have added the **Table R2** as the **Supplementary Table 3** and also added specific explanation to the revised manuscript as follows:

Lines 190-193: "Since the enhanced variability of EAJ reaches 2.77 m s⁻¹ averaged from all datasets (see **Fig. 1c** and **Supplementary Table 1**), our primary investigation suggests that the SCA contributes $81.1 \pm 2.9\%$ of the enhanced variability (2.25 m s⁻¹ relative to 2.77 m s⁻¹, **Supplementary Table 3**).

Table R2 | Contribution of SCA on the enhanced variability of EAJ based on seven datasets. Contribution of SCA in SD of EAJ is calculated as the SD difference between the original SD and that after linearly removing the signal of SCA. The mean values are shown in red and bold.

Datasets	Change in SD of EAJ from Table 1 (P2 – P1; m s ⁻¹)	Contribution of SCA in SD of EAJ (m s ⁻¹)	Percentage
ERA-5	2.71	2.09	77.1%
JRA-55	2.77	2.24	80.9%
ERA-I	2.81	2.17	77.2%
NCEP-NCAR	2.87	2.18	76.0%
NCEP/DOE	2.91	2.36	81.1%
MERRA-2	2.74	2.41	88.0%
CFSR	2.61	2.29	87.7%
Mean	2.77 ± 0.10	2.248 ± 0.11 (Rounded to 2.25 ± 0.11)	81.1% ± 2.9%

7. Lines 180-181. The zonal wind responses over other regions (Extended Data Fig. 6c) are different from those based on reanalyses (Extended Data Fig. 5b). Need to make some comments on this aspect.

Response: We would like to apologize before proceeding with our response. Because in the old Extended Data Fig. 6, we made a mistake in calculating the simulated circulation response with the average of August and September, rather than the mean of July and August. Now we have corrected it as the new **Supplementary Fig. 10** (i.e., the following **Fig. R4**). And we have also double-checked all analysis codes in this work and are confident that all the other results have been correctly processed.

After correction, the simulated results can generally capture the enhanced EAJ responses (**Fig. R4**). However, the zonal wind responses over other regions still show discrepancies with the observed counterpart shown in **Supplementary Fig. 9** (i.e., the old Extended Data Fig. 5). Such discrepancies between observation and

model simulation can be attributed to several aspects, including the bias within the model (**Fig. R5**), which have been added into the revised manuscript as follows:

Line 216-228: “Given that the simulated circulation responses over other regions show discrepancies with the observed counterpart (**Supplementary Fig. 9**), our conclusion on the simulated result should be interpreted carefully. Such discrepancies can be attributed to the following several aspects. For instance, the mean state in the model exhibits a weaker EAJ but stronger westerlies over higher latitude (**Supplementary Fig.11**), which may cause the ray path that originates from the North Atlantic region differs somewhat from the observed one^{44,45}. In addition, the Rossby wave response is highly variable with time, which makes it difficult to capture the observed wave structure in model⁴⁶. Finally, the model’s resolution, simplifications, assumptions, and parameterizations limit its capability to represent complex climate features⁴⁷. Despite the discrepancies between the simulated wave train pattern and its observed counterpart, this result still strongly suggests that the Northernmost Atlantic around Iceland is indeed a key region generating wavetrain to the East Asia.”

Reference:

44. Ding, Q. et al. Tropical forcing of the recent rapid Arctic warming in northeastern Canada and Greenland. *Nature* 509, 209–212 (2014).
45. Li, R. K. K., Woollings, T., O’Reilly, C. & Scaife, A. A. Effect of the North Pacific Tropospheric Waveguide on the Fidelity of Model El Niño Teleconnections. *J. Clim.* 33, 5223–5237 (2020).
46. White, R. H., Kornhuber, K., Martius, O. & Wirth, V. From Atmospheric Waves to Heatwaves: A Waveguide Perspective for Understanding and Predicting Concurrent, Persistent, and Extreme Extratropical Weather. *Bull. Am. Meteorol. Soc.* 103, E923–E935 (2022).
47. Neale, R. B. et al. The Mean Climate of the Community Atmosphere Model (CAM4) in Forced SST and Fully Coupled Experiments. *J. Clim.* 26, 5150–5168 (2013).

Fig. R4 | Simulated atmospheric response to SSTA forcing. a, High summer (JA) SST anomalies (shading; units: K) near the Iceland as the oceanic forcing in CAM4 experiment. b–c, Simulated atmospheric response in (b) H300 and (c) U200 to prescribed oceanic forcing with 95% significance stippled. The red box outlines the region used for defining the EAJ index. The blue line in c indicates the jet axis.

Fig. R5 | Mean state bias of CAM4 model. **a**, the mean state of U200 (units: m s^{-1}) in CAM4 model in CTRL run as the climatology of model. **b**, climatology of U200 (units: m s^{-1}) during 1979-2018, obtained from ERA5 data. **c**, difference between climatology of CAM4 and ERA5, with the corresponding jet stream axis (blue for CAM4 and red for ERA5).

8. Line 337-338. Why do authors use two different longitude ranges?

Response: Thank you for pointing out this issue. Now we have used the same longitude ranges (i.e., 80° – 130° E) to calculate the leading modes of zonal wind anomalies in the vertical profile of (15° – 65° N, 1000-hPa–100-hPa; averaged from 80° – 130° E), and added the corresponding **Supplementary Fig. 4** (i.e., the following **Fig. R6**) in the revised **Supporting Information**.

We have also unified the longitudinal range in **Supplementary Fig. 3b, d** to 80° – 130° E.

Fig. R6 | Shifting of leading mode of the zonal wind in vertical profile. Same as Fig. 3a, b, and e but the EOF analysis is performed using the meridional section of zonal wind (i.e., 15°–65°N, 1000-hPa–100-hPa; averaged from 80°E to 130°E). The blue contours indicate the climatology for zonal wind in 15 and 25 m s⁻¹.

9. Fig. 1c. Linear trend for WAJ is -0.08 m s^{-1} per decade. See major comment 5 and specific comment 15.

Response: Corrected. Please see response to above major comment 5.

10. Fig. 2. Are these regressions or composites of heat wave frequency and drought conditions? Clarify.

Response: All sub-figures of Fig. 2 represent the regression maps against the normalized EAJ index. We have clarified the expression in the figure legend.

11. Fig. 3. Use the same latitude range in panel b and d.

Response: Done.

12. Line 558. “natural state TPH”, what is it? Do you mean neutral year of EAJ or climatology?

Response: The term “natural state of TPH” refers to “the climatology of TPH”.
Now it has been clarified in the figure legend of **Supplementary Fig. 3**.

13. Extended Data Fig. 5. Please plot climatological jet axis in panel b. The sub-label “Geopotential Height” in panel b is not correct.

Response: Done as suggestion and corrected. We have added the climatological jet axis (red line) and corrected the sub-label in panel **b** for **Supplementary Fig. 9** in the revised **Supporting Information** (i.e., as shown in the **Fig. R7** below).

Fig. R7 | The revised Supplementary Fig. 9b.

14. Extended Data Fig. 6. Please plot model climatological jet axis in panel c.

Response: Done, see the new **Supplementary Fig.10** (i.e., above **Fig. R4**).

15. Extended Data Table 1. The WAJ trends range from -0.65 to -0.98 m s⁻¹ per decade.
See major comment 5 and specific comment 9.

Response: Corrected, thanks for your careful feedback. Please see the response to above major comment 5.

Typos

1. Line 370. “Ice Land” to “Iceland”
2. Line 602, “Ice Land” to “Iceland”

Response: Corrected, thanks.

Response to reviewer #2:

This study reported an unprecedented strong variability of EAJ intensity since the late-1990s, which is closely associated with large-range climate extremes in East and South Asia, and its driving factor of the northernmost Atlantic SSTA. This study also revealed that the persistent SSTA signals of EAJ variability the late-1990s are initiated by the Scandinavian pattern in February. The authors have demonstrated in detail the intrinsic correlation between the Scandinavian pattern in February, the persistent North Atlantic SSTA anomalies and the EAJ variability in high summer, under the hypothesis of "cross seasonal coupled oceanic atmospheric bridge". The argumentation and the conclusions are convincing. This study tells a complete and interesting story, which has important reference value. It should be considered for publication after the following revisions.

Response: Thank you for the positively supportive comments. The quality of the manuscript has been greatly improved based on your comments. Please find below a detailed point-by-point response to all comments.

Major comments:

1. Although I am convinced that the Northernmost Atlantic around Iceland is indeed a key region of the SSTA forcing of generating the strong EAJ variability, the simulated wave train response somehow differs from its observed counterpart. How to justify it?

Response: Agree. Here we would like to apologize before proceeding with our response. Because in the old Extended Data Fig. 6, we made a mistake in calculating the simulated circulation response with the average of August and September, rather than the mean of July and August. Now we have corrected it as the new **Supplementary Fig. 10** (i.e., the following **Fig. R1**). And we have also double-checked all analysis codes in this work and are confident that all the other results have been correctly processed.

After correction, the simulated results can also generally capture the enhanced EAJ responses (**Fig. R1c**). However, the simulated wave train response still somehow differs from its observed counterpart shown in **Supplementary Fig. 9** (i.e., the old Extended Data Fig. 5). Such discrepancies between observation and simulation can be attributed to several aspects, including the bias within the model (**Fig. R2**), which have been added into the revised manuscript as follows:

Line 231-242: “Given that the simulated circulation responses over other regions show discrepancies with the observed counterpart (**Supplementary Fig. 9**), our conclusion on the simulated result should be interpreted carefully. Such discrepancies can be attributed to the following several aspects. For instance, the mean state in the model exhibits a weaker EAJ but stronger westerlies over higher latitude (**Supplementary Fig. 11**), which may cause the ray path that originates from the North Atlantic region differs somewhat from the observed one^{44,45}. In addition, the Rossby wave response is highly variable with time, which makes it difficult to capture the observed wave structure in model⁴⁶. Finally, the model’s resolution, simplifications, assumptions, and parameterizations limit its capability to represent complex climate features⁴⁷. Despite the discrepancies between the simulated wave train pattern and its observed counterpart, this result still strongly suggests that the Northernmost Atlantic around Iceland is indeed a key region generating wavetrain to the East Asia.”

Reference:

48. Ding, Q. et al. Tropical forcing of the recent rapid Arctic warming in northeastern Canada and Greenland. *Nature* 509, 209–212 (2014).
49. Li, R. K. K., Woollings, T., O’Reilly, C. & Scaife, A. A. Effect of the North Pacific Tropospheric Waveguide on the Fidelity of Model El Niño Teleconnections. *J. Clim.* 33, 5223–5237 (2020).
50. White, R. H., Kornhuber, K., Martius, O. & Wirth, V. From Atmospheric Waves to Heatwaves: A Waveguide Perspective for Understanding and Predicting Concurrent, Persistent, and Extreme Extratropical Weather. *Bull. Am. Meteorol. Soc.* 103, E923–E935 (2022).
51. Neale, R. B. et al. The Mean Climate of the Community Atmosphere Model (CAM4) in

Fig. R1 | Simulated atmospheric response to SSTA forcing. a, High summer (JA) SST anomalies (shading; units: K) near the Iceland as the oceanic forcing in CAM4 experiment. **b–c**, Simulated atmospheric response in **(b)** H300 and **(c)** U200 to prescribed oceanic forcing with 95% significance stippled. The red box outlines the region used for defining the EAJ index. The blue line in **c** indicates the jet axis.

Fig. R2 | Mean state bias of CAM4 model. **a**, the mean state of U200 (units: m s^{-1}) in CAM4 model in CTRL run as the climatology of model. **b**, climatology of U200 (units: m s^{-1}) during 1979-2018, obtained from ERA5 data. **c**, difference between climatology of CAM4 and ERA5, with the corresponding jet stream axis (blue for CAM4 and red for ERA5).

2. The strong EAJ variability since the late-1990s was initiated by the Scandinavian pattern in February instead of early spring. The February pattern is distinct and quite different from the typical winter and spring patterns. The authors should be with caution.

Response: Agree, thank you for pointing out this issue. According to the study of Bueh and Nakamura (2007), the SCA pattern in February is indeed distinct and quite different from the other winter and spring months in both spatial distribution and mechanisms (**Fig. R3**). Thus, we have replaced the “early spring” as “February” to create a more concise expression. Besides, we have added the following

discussions to the revised manuscript.

Line 183-186: “Of note is that, here the strong EAJ variability since the late-1990s is initiated by the SCA in February instead of early spring since the February pattern is distinct and quite different from the typical winter and spring patterns (see the Fig.1 in Ref⁴³).”

[REDACTED]

Fig. R3 | “Maps of the linear regression coefficient between local stream-function-like height anomaly at the 300 mb level and the PC time series of the SCA mode.”
Adopted from Bueh and Nakamura (2007).

Reference:

43. Bueh, C. & Nakamura, H. Scandinavian pattern and its climatic impact. *Q. J. R. Meteorolog. Soc.* 133, 2117–2131 (2007).

Minor concerns

1. L123-124: which is identical to EAJ in both spatial distribution (Extended Data Fig. 3a)—the figure legend is not correct.

Response: Corrected. It should be the Extended Data Fig. 1a in the original manuscript. Now it is the new **Supplementary Fig. 3a (Line 144)**.

2. L131-134: “A stronger EAJ, with strengthened anticyclonic wind shear in its south flank, can induce the eastward-extended Tibet Plateau High in the upper-troposphere, enhanced western North Pacific subtropical high in the mid-

troposphere and significant warming near surface (Extended Data Fig. 1c, e) --- the authors should be careful whether there is a causal relationship among them.

Response: Thank you for pointing out this issue. Now we have rewritten this sentence and add relevant discussions to the revised manuscript as follows.

Line:154-162: “Namely, a stronger EAJ, with strengthened anticyclonic wind shear in its south flank, favors the eastward-extended Tibet Plateau High in the upper-troposphere and the westward-enhanced western North Pacific subtropical high (WNPSH) in the mid-troposphere (**Supplementary Fig. 3e**), thereby together contributing to strong downdraft *in-stu* (**Supplementary Fig. 3d**) extending through the depth of the troposphere³⁹. Meanwhile, given the enhanced WNPSH, more water vapor is transported to north of $\sim 32^{\circ}\text{N}$, leading to more convective precipitation but rainfall deficit to south of $\sim 32^{\circ}\text{N}$ (**Fig. 3d**). Besides, the enhanced WNPSH warms subtropical China and the East China Sea via adiabatic subsidence and more incoming solar radiation (**Supplementary Fig. 3c**).”

Reference:

39. Zhang, P., Yang, S. & Kousky, V. E. South Asian high and Asian-Pacific-American climate teleconnection. *Adv. Atmos. Sci.* 22, 915–923 (2005).

3. L156-157: “with almost identical spatial pattern (Extended Data Fig. 2a; pattern correlation coefficient reaches 0.99)--- it needs to be clarified.

Response: Clarified. The following **Fig. R4** show the correlation and regression maps of February geopotential height anomalies against EAJ index and SCA index at the upper-, middle- and lower-troposphere (i.e., H200, H500 and H850). It is clear that they are highly identical throughout the troposphere, albeit with weaker magnitude for EAJ. We further calculate their pattern correlations between circulation patterns obtained from EAJ and SCA over the region 25°N – $85^{\circ}\text{N}/65^{\circ}\text{W}$ – 70°E (i.e., blue box in **Fig. R4**). As expected, the pattern correlation in each pressure-level, regardless of the use of regression maps or correlation maps,

uniformly exceeds 0.98 (see the following **Table R1**). We have rewritten this sentence in revised manuscript as follows.

Lines 180-183: “As expected, the SCA index is highly correlated with EAJ index ($R = 0.84$, $p < 0.001$; **Fig. 4b**); and the corresponding pattern correlation coefficient exceeds 0.98 in each pressure-level over the region 25°N – $85^{\circ}\text{N}/65^{\circ}\text{W}$ – 70°E (see **Supplementary Table 4** for details).”

Fig. R4 (i.e., the new Supplementary Fig. 5) | Comparing circulation patterns of EAJ and SCA at February. a–c, Correlation (shading) and Regression maps (contour; units: gpm) of (a) H200, (b) H500 and (c) H850 in February associated with EAJ index for the post-1999 period. d–f, Same as a–c, except for the February SCA index. Blue boxes outline the same region (25°N – 85°N , 65°W – 70°E) used for calculating their pattern correlations, as shown in the following **Supplementary Table 4.**

Table R1 | Pattern correlations between the circulation modes shown in Supplementary Fig. 5. Pattern correlations (see methods) are calculated for SCA pattern in the same region (25°N–85°N, 65°W–70°E) between that obtained from EAJ index and SCA index, at 200 hPa, 500 hPa and 850 hPa, respectively.

Pattern Correlation	H200	H500	H850
Correlation map	0.991	0.980	0.986
Regression map	0.993	0.988	0.990

REVIEWERS' COMMENTS

Reviewer #1 (Remarks to the Author):

Review's comments for the revised paper (NCOMMS-23-57943-A), entitled "Atlantic origin of the sharply increasing Asian westerly jet variability", submitted to Nature Communications

Recommendation: Minor revision

Most of my comments on the early version of the paper have been addressed in a satisfactorily manner in the revised version and the corresponding responses. However, there are still some specific comments listed below that need to be addressed. The paper is, therefore, acceptable for publication after a minor revision.

Specific comments

1. The study is about interannual variability in summer. This is better to be explicit in the title. Authors only need to add two words such as "Asian summer westerly jet interannual variability", which would not make title too long.
2. Line 42. "record-breaking jet stream", in what aspect? Rephrase.
3. Lines 105-109. These statements seem not true given the fact that Fig. 1c and Supplementary Table 1 show significant decreasing trends of WAJ and EAJ. Clarify and revise.
4. Line 119. Why does "the weakened polar jet (Supplementary Fig. 3a) favors more synoptic disturbances"?
5. Line 211. "Genergal" to "General".
6. Line 254. The study is about EAJ in summer. Please add "summer" in this sentence.

Reviewer #2 (Remarks to the Author):

The authors have addressed my questions and comments properly in the revised manuscript. This study is worthy for publication.

Response to Reviewers' Comments

Response to reviewer #1:

Review's comments for the revised paper (NCOMMS-23-57943-A), entitled "Atlantic origin of the sharply increasing Asian westerly jet variability", submitted to Nature Communications

Recommendation: Minor revision

Most of my comments on the early version of the paper have been addressed in a satisfactory manner in the revised version and the corresponding responses.

However, there are still some specific comments listed below that need to be addressed. The paper is, therefore, acceptable for publication after a minor revision.

Response: Thank you for the positively supportive comments. In this revision, we have further refined the explicit explanations. Please find below a detailed point-by-point response to all comments.

Specific comments

1. The study is about interannual variability in summer. This is better to be explicit in the title. Authors only need to add two words such as "Asian summer westerly jet interannual variability", which would not make title too long.

Response: Done as suggestion. And according to the suggestion from the Author Checklist of this journal, the title has been revised as "Atlantic origin of the increasing Asian westerly jet interannual variability".

2. Line 42. "record-breaking jet stream", in what aspect? Rephrase.

Response: Thank you for pointing out this issue. The record-breaking jet stream in summer 2020 is referred to the strongest Asian subtropical westerly jet since 1979. We have rewritten this sentence as follows:

Line 42-44: "Summer 2020 saw an extreme rainfall over the middle and lower reaches of the Yangtze River Valley, triggered by a record-breaking Asian subtropical jet stream in its intensity¹⁰⁻¹²."

3. Lines 105-109. These statements seem not true given the fact that Fig. 1c and Supplementary Table 1 show significant decreasing trends of WAJ and EAJ. Clarify and revise.

Response: Thank you for pointing out this issue. We admit that there exists decreasing trend for WAJ and EAJ, while both are insignificant at 0.05 confidence level based on the t -test. For WAJ, the p values for its linear trends range from 0.054 to 0.159 among seven datasets; for EAJ, the p values range from 0.095 to 0.286. We have clarified the confidence level in this sentence as follows:

Line 106-107: “In addition, there is no statistically significant weakening trend in strength over the majority of Eurasian jet ($p > 0.05$ for all three indices in all datasets; **Fig. 1c and Supplementary Table 1**) ...”

4. Line 119. Why does “the weakened polar jet (Supplementary Fig. 3a) favors more synoptic disturbances”?

Response: Firstly, the weakened jet stream, as the background mean flow, would slow down the propagation of synoptic disturbance.

Secondly, the weakened jet stream would cause larger amplitude of synoptic Rossby wave and prompt the high-frequency wave activity (**Fig. R1**).

We have rewritten this sentence to make a more concise expression, as follows:

Line 120-123: “Acting as the background condition for synoptic disturbance, the weakened polar jet (**Supplementary Fig. 3a**) would impede the propagation of synoptic Rossby waves as well as larger amplitude²², enhancing synoptic scale wave activity. The consequently enhanced meridional eddy heat mixing^{31,32} would lead to warming over the high-latitude.”

[REDACTED]

Fig. R1. Schematic for the jet stream meandering with respect to zonal jet speed. Arrows depict the mean zonal jet. Red and blue shadings describe the northward and southward excursions of the westerly jet, respectively. (a) A fast westerly jet acts as a “mixing barrier” for cross-jet transport. (b) A slow westerly jet allows enhanced eddy mixing in latitude. Adopted from Chen et al. (2022)

Chen, G., Nie, Y. & Zhang, Y. Jet Stream Meandering in the Northern Hemisphere Winter: An Advection–Diffusion Perspective. *J. Clim.* 35, 2055–2073 (2022).

5. Line 211. “Genergal” to “General”.

Response: Corrected, thanks.

6. Line 254. The study is about EAJ in summer. Please add “summer” in this sentence.

Response: Done, thanks.

Response to reviewer #2:

The authors have addressed my questions and comments properly in the revised manuscript. This study is worthy for publication.

Response: Thank you for your valuable time and effort in reviewing our manuscript. We sincerely appreciate all helpful comments and suggestions which help us to improve the quality of our manuscripts.